# The effects of video racing games on risk-taking in consideration of the game experience

Ewelina Stollberg *[ORCID]*[◔], Klaus W. Lange[◔]

Institute of Psychology, University of Regensburg, Regensburg, Germany

◔ These authors contributed equally to this work.
* ewelina.stollberg@ur.de

## Abstract

In the present study, we attempted to investigate whether it is the game experience that underlies processes leading to increased risk-taking while playing video racing games (VRGs). The aim of the study was to examine the indirect effect of playing VRGs on risky driving behaviour through various dimensions of the game experience. Thus, we examined the subjective experience of participants playing various VRGs and questioned whether this influenced subsequent driving behaviour. The results of the present study show that aspects of the playing experience, in particular "flow" and "competence", appear to be important in the regulation of risk appetite. It can be assumed that, rather than the mere playing of VRGs, the type of game experience during play is determinant for subsequent risk propensity while driving.

## Introduction

Digital media have established themselves as an integral part of our society. Given the widespread use of devices such as smartphones, laptops, etc., they have become a major factor in the leisure activities of children and adolescents. Engagement with the digital world starts at increasingly younger ages. The term "digital natives" has been coined to refer to a generation that has grown up with computers and the internet [1]. The consumption of video games plays a particularly important role among this age group. According to a 2014 BITKOM study on media consumption of children and young people in Germany [2], 93% of children aged 10 to 18 years regularly play computer and video games. On average, these children spend approximately 104 minutes a day playing these games, with boys playing longer than girls. Nearly one of every five children aged 16 to 18 stated that they spend an average of three hours or more a day using computers, gaming consoles, and smartphones, and over a third of 10-to 18-year-olds stated that they prefer to play alone. This development raises the question of how the consumption of video games affects children's behaviour and, more especially, the development of the irpersonalities. A common problem in this field of research is the creation of dichotomous categories of "good" and "bad". The complexity of the games and their impact on motivation, feelings, and social interaction are often underestimated. The most popular and successful

**Data Availability Statement:** All relevant data are within the manuscript and its Supporting Information files.

**Funding:** The authors received no specific funding for this work.

**Competing interests:** The authors have declared that no competing interests exist.

games today are characterised by innovation and continuous development in terms of the playing experience. The variety of the games on offer is enormous and cannot be limited to one or two genres. More classic games, such as racing games including "Need for Speed", party games such as "Mario Kart", and sport games such as "FIFA" compete increasingly with games involving role-playing and strategic elements.

To date, a considerable amount of research has been undertaken investigating the links between media consumption, including that of video games, and elevated levels of violence and aggressive behaviours, cognition, emotions, and arousal [3–5]. However, the effects of risk-glorifying media content on subsequent engagement in risky behaviours has, as yet, received less attention. One of the few, albeit extensive studies is the meta-analysis by Fischer and colleagues [6], which looked at 88 studies including both correlative and experimental designs. The consequences of consumption of various media formats were investigated. Dependent variables included risk-taking behaviours (such as tobacco consumption, alcohol consumption, risk-taking while driving, risk-prone sexual behaviours) as well as risk-positive thoughts, emotions, and arousal. The results showed a significant positive correlation between the consumption of risk-glorifying media and the tendency to engage in risk-tolerant behaviours. There was also a significant positive correlation between the use of risk-glorifying media and risk-positive cognition, attitudes, emotions, and arousal. However, the overall effect was deemed only minimal to moderate in size [6]. Research to further elucidate the consequences of consuming risk-glorifying media content is needed. Risk-promoting video racing games (VRGs) are particularly relevant since they are a common pastime among adolescents [7, 8] and young adults today [9], and there are still relatively few studies on this subject. This type of video game is characterised by aggressive driving, the risks of which are downplayed, and the rewarding of risk-taking. Current VRGs are frequently designed in such a way that the player, playing from the driver's perspective, races at high speeds through cities and landscapes, provoking accidents, and performing foolish overtaking manoeuvres [10]. The few studies on the effects of risk-glorifying VRGs indicate that these games may increase players' risk tolerance at the levels of cognition, emotion and arousal both while playing and in real-world driving [7, 10–12].

According to Wilde [13], the level of risk people are willing to accept is dependent on both the cost and benefit of the risky behaviour compared to safer choices. Wilde describes the risk level with maximum benefit as the "target level". People compare the perceived level of risk with the target level at all times and try to minimise this difference by adjusting their behaviour. This adjustment, which also depends on the ability to make decisions and on driving ability, changes the probability of accidents. These factors together with information from friends, family or the media about accidents, influences the perceived level of risk in a particular situation. The perceived risk level is also influenced by the person's perceptual skills in general [13]. This theory is interesting in the context of the effects of risk-glorifying video games on behaviour since it explains one of the basic mechanisms underlying the readiness to assume risk on the road. Several investigations have shown that risk-glorifying video games lead to an increased propensity to take risks while driving [10, 11].

The General Learning Model (GLM) [14] is an extension of the General Aggression Model (GAM), devised earlier by Bushman and Anderson [15]. As in the original model, the GLM integrates various socio-cognitive approaches, including social learning [16], the cognitive neo-association model [17], social information processing [18], and script theory [19]. The GLM can also be applied to the playing of risk-glorifying video games and their impact on subsequent behavioural intentions. The individual input variables that can influence the internal state while playing risk-glorifying video games include prior experience with risk-promoting video games, certain personality traits such as anxiety or thrill-seeking, as well as attitudes and

habits in regard to risky actions. The playing of risk-promoting VRGs requiring the risky and reckless driving of a car via a joystick or other controller. With the latest technology, many of these games are graphically realistic, creating a perception of driving in the real world and racing cars at high speed with no negative consequences. Many of the games feature realistic engine noise and music with fast rhythms. These personal and situational input variables influence the internal state by priming risk-seeking cognition, which, in turn, has an effect on affect and the state of excitement or vice versa. Frequent playing of such games can result in long-term changes in behaviour [11, 14].

In a series of studies, Fischer and colleagues [11] demonstrated that the consumption of VRGs is associated with an increased risk tolerance in road traffic. In their correlation study, they combined four variables to create a risk scale for real-world driving: (1) competitive road traffic behaviour, (2) the need to show off, (3) cautious driving behaviour and (4) the number of reported accidents, each recorded using specially constructed items. In addition, the participants were asked to indicate on a scale from "0" (never) to "5" (daily) how often they played VRGs (e.g. "Need for Speed") and other video games. A hierarchical regression analysis was then used to evaluate the data. The results confirmed the connection between racing games and increased risk-taking on public streets. In the second experimental study, the authors examined whether racing game consumption resulted in greater accessibility of risk-related cognition and affect. The participants played either one of three VRGs ("Burnout", "Midnight Racer", or "Need for Speed") or one of three neutral games ("Tak", "Crash Bandicoot", or "FIFA 2005") for 20 minutes. In all three racing games, participants competed against computer-generated opponents and had to break traffic rules in order to win. The accessibility of risk-favouring cognition was assessed with an implicit measure, the "homonymous decision task". Concepts had to be defined based on given words that had both a risk-oriented and a non-risk-oriented meaning. Arousal was measured according to various adjectives on a scale from "0" to "10". Affect was recorded with the "Positive and Negative Affect Schedule" (PANAS) [20]. Compared to the control group, those who played racing games found it easier to access risk-related cognition and stronger arousal. No significant group differences were observed with respect to affect. The third study sought to determine whether the players of VRGs showed a greater willingness to take risks in typical traffic situations. The authors expected the results of the second study to be replicated. After playing, the subjects completed the "Vienna Risk-Taking Test Traffic" (WRBTV). The accessibility of risk-favouring cognition was again measured with the "homonymous decision task". The assumptions of the authors were confirmed by the results. VRGs seem to have a potential effect on a wide range of measures, including risk tolerance and cognition.

Silberman [21] examined the relationship between the consumption of various forms of different and the driving habits of young people. The author's initial assumption was that the more young people use media such as risk-glorifying VRGs, the more likely they will be to pursue similar actions in reality and engage in dangerous driving habits. Young adults (aged 17 to 25) were asked how often they played the following risk-promoting games: "Grand Theft Auto", "Need for Speed" or "Burn-out" on a scale from "1" ("never") to "4" ("more than once a week"). An overall score was then calculated based on their responses. To record their driving habits, they were asked to respond to 33 items, using a five-level Likert scale, assessing level of adherence to traffic rules and care taken while driving. The consumption of video games involving risky driving manoeuvres proved to be a good predictor of the respondents' driving habits. This supports the view that VRGs can influence the real world behaviour of those playing them?

Deng and colleagues ([22]—experiment 1) used a student sample to determine whether VRGs influence risk-favourable attitudes. The role of personality traits was also investigated. The Eysenck Personality Profile and the WRBTV were used to measure respondents' personality

traits and risk-taking while driving. The subjects of the experimental group were asked to race against the clock in a circuit racing game lasting 20 minutes. The control group played Poker Solitaire. The subjects who played the racing game tended more towards risky behaviours in road traffic scenarios than participants who had played neutral games. A lust for adventure as a personality characteristic also showed a significantly positive correlation with risk propensity. The more adventurous respondents were, the more likely they were to take risks on the road.

The studies detailed below demonstrate that video games may have both short- and long-term effects on risk-tolerance, cognition, and arousal. The important question as to the source of this effect at a psychological level remains unclarified.

According to the GLM, personal and situational factors act together to influence an individual's internal state, potentially inducing changes at the levels of cognition, emotion, and arousal [4, 14]. The player experience is an important aspect of these game-induced experiential changes. Research in a variety of disciplines is focusing on the player experience, representing an increase in research on the part of the game industry, which sees this as an opportunity to improve games by designing them to give players the optimal playing experience with maximum enjoyment (for detail see [23]). In the field of media psychology, there has also been an increase in the number of publications addressing the player experience in video games (for detail see [24–26]). In light of the fact that so much of people's lives is taken up with video games [7, 8], it is becoming increasingly important to ascertain what effect certain video games have on players' lives, thoughts, emotions and level of excitement, and whether the player experience differs between different video games. This knowledge could help explain why some video games lead to changes in behaviour and may help us understand the underlying processes.

Among the most important dimensions of the player experience discussed in the current research are the level of *immersion* [25–31] and the level of *presence*, where the virtual world is perceived to the same degree as the real world [31, 32]. Another aspect is *flow*, which describes an intrinsically motivating state of pleasure [25, 33–37]. Added to this is the dimension of *competence* or *self-efficacy*, i.e. the subjective confidence that a behaviour can successfully be carried out in order to achieve a specific goal [38–42]. However, it is also apparent that the players' experiences of *competence* and *flow* are strongly interdependent [43]. Players experience *flow* mostly when there is a balance between their ability and the difficulty level [43]. *Tension*, which can be seen as a positive experience in the current literature [44–46], is, in this context, seen as a negative emotional state [47–51]. Finally, there is *challenge* or *competition*, representing the subjective sense of being challenged [43, 52–54].

The individual player experience can be seen as an important part of the player's internal state, as described in the GLM. As a result, players' experiences while playing video games may help explain changes in behaviour after repeated learning processes.

The subjective experience of people playing risk-glorifying VRGs has so far received little attention. Few studies examining the effect of this kind of video game on the emotional state of players have been conducted [10, 11]. Since the processes through which an increased risk-tolerance in real-world driving may result from the playing of VRGs remain unexplored, this research gap should be closed. In order to discover the cause of increased risk-tolerance following the playing of certain video games, a detailed examination of which type of player experience is associated with which type of risk-promoting video game would be of interest.

## Materials and methods

### Participants and study design

One hundred and twenty participants, primarily students from the University of Regensburg (57 men and 63 women; M = 23.59, SD = 3.33) aged between 18 and 34 years participated in

this study. The age range of male participants was 19–34 years (M = 24.36, SD = 3.53) and of female participants 18–30 years (M = 22.92, SD = 3.01). The participants were randomly divided into one of two conditions, each comprising 60 persons. All played either risk-glorifying VRGs, with the experimental group playing *drive'em up games*, or *rule-compliant VRGs*, played by the control group. In both conditions, the participants played one of the games for 25 minutes on three consecutive days. The *drive'em up games* group comprised 30 men and 30 women, and the *racing games* group 27 men and 33 women. The age of the participants in the experimental group had a mean value of M = 23.64 (SD = 2.85, age range = 19–30 years) and the age in the control condition was between 18 and 34 years (M = 23.54, SD = 3.78).

Participants were excluded from participation for the following reasons: severe sensory impairments (especially a reduced sense of hearing or sight), addictive disorders (excluding nicotine), seizure disorders, neurological or psychiatric disorders or medication with drugs affecting the central nervous system. Participants were rewarded by participation in a lottery of three times 100 Euro.

The study was based on a one factorial between-subject design with the two-level led factor "game condition". The study received ethical approval from the University of Regensburg Ethics Committee.

## Procedure

The participants were recruited via a notice board at the University of Regensburg. Prior to the commencement of the experiment, participants were required to sign a study information sheet and to provide written consent to participation. At this stage, they were not informed of the question to be determined in the investigation. On the first day of the experiment, which took place in the game-laboratory at the University of Regensburg, the participants were given a written instruction to the corresponding video racing games. This was followed by a 5-minute practice period, during which the participants familiarised themselves with the game. During the following 20 minutes, the experimental group played one of three *drive'em up games* and the control group played one of three *racing games*. The order of the games was randomised in both groups. Participants playing the risk-glorifying VRGs were briefed that the aim of the game is to drive as fast as possible and win the game. Participants in the control condition were informed that the goal is to win the game, but that driving behaviour norms should be followed and collisions avoided. The car could be damaged by any collisions, which would negatively affect the race. On completion of the game, participants were asked to fill in the "Game Experience Questionnaire" (GEQ), German Version [26]. On the second and third day of the experiment, the procedure was repeated (a 5-minute training period, followed by 20minutes of VRG playing and completion of the GEQ). After filling in the GEQ on the third day, participants performed the WRBTV, which took approximately 20 minutes. Finally, the participants were informed of the purpose of the experiment.

## Materials

The following *drive'em up games* were selected for the experimental condition: "Need for Speed—Hot Pursuit", "Blur" and "Motor storm—Pacific Rift". These VRGs were deemed appropriate, since they reward players for fast driving and violation of road traffic rules. For example, driving on the wrong side of the road, tailgating, and provoking near-accidents are rewarded. The following rule-compliant VRGs were chosen for the control condition: "Need for Speed Shift 2 unleashed", "Formula 1 2010" and "Grand Turismo 5". All three rule-compliant games require a driving style in accordance with the norm, since they use a realistic

damage model. In this kind of game, accidents with rivals lead to damage to the player's own car, which negatively affects subsequent driving performance.

The games were selected according to their popularity in internet forums (e.g. www. gamespot.com and ww.metacritic.com) and good sales figures (www.vgchartz.com). All participants played in the third-person perspective. The cars and the race circuits were selected by the investigators and were identical for all participants.

All games were played on a Sony PlayStation 3 platform and a 50-inch TV flat-screen with 5.1 dolby-surround-systems.

## Measures

**Game Experience Questionnaire (GEQ).** The various dimensions of the game experience were assessed using the "Game Experience Questionnaire" (GEQ), German Version [26]. The GEQ is divided into seven subscales: *immersion* (Cronbach's α = .891 and factor loadings ranging from .587 to .847), *flow* (Cronbach's α = .866 and factor loadings ranging from .614 to .821), *competence* (Cronbach's α = .826 and factor loadings ranging from .474 to .755), *tension* (Cronbach's α = .811 and factor loadings ranging from .606 to .704), *challenge* (Cronbach's α = .745 and factor loadings ranging from .605 to .348), *positive affect* (Cronbach's α = .797 and factor loadings ranging from .430 to .728) and *negative affect* (Cronbach's α = .712, all factor loadings rather low). Except for the scale *immersion*, which consists of six items, each dimension has five items. Example statements are, "I felt that I could explore things" (*immersion*), "I forgot everything around me" (*flow*), "I felt skillful" (*competence*), "I felt irritable" (*tension*), "I felt stimulated" (*challenge*), "I enjoyed it" (*positive affect*), and, "I felt bored" (*negative affect*). Responses were given on a Likert scale from 0 = "not at all" to 4 = "extraordinarily" [55]. The seven factors solution proved to be the statistically most meaningful and explained 52% of variance [55].

In order to compare the differences between the video-racing groups in the GEQ, mean values for each of the seven subscales were calculated. Since the GEQ was applied three times per subject (after playing each of the three different video games), the game-specific subscale scores were collapsed (by summing) within each subject to form a total (sub)scale score reflective of respective game experience across the different games.

**Vienna Risk-Taking Test Traffic (WRBTV).** Risk prone driving behaviour was examined with the "Vienna Risk-Taking Test Traffic" (WRBTV) [56]. This method is used to measure the subjectively accepted level of risk in traffic situations according to the risk homeostasis theory [57]. This test is employed in traffic psychology and is based on reaction times. Written instructions on test procedure, displayed on a computer screen, were read by participants, who then viewed a video depicting 23 dangerous road traffic situations. Each situation, which was preceded by a written description displayed on the screen, was viewed twice. The first time, respondents were asked to observe carefully and the second time to press a green button on the response panel at the point they would no longer perform the driving maneuver (e.g. overtaking). For statistical analysis, the mean value of the time delay before pressing the button (averaged a cross all 23 driving situations) is used. The longer the latency, the more willing to take risks the participant is judged. According to calculations reported by Hergovich and colleagues [57], the reliability of the WRBTV, as indicated by its internal consistency, is good (Cronbach's α = .924).

## Statistical analysis

Mediation analyses were conducted with a parallel multiple mediator model. Mediation was analysed by testing the significance of the indirect effect of playing different types of VRGs (X)

on risky driving behaviour (Y) through several dimensions of the game experience (M). In the path modeling framework, indirect effects can be conceptualised as the product of the effect of the X on M ($a$ path) and the effect of M on Y ($b$ path) (see Fig 1).

The total effect of risky driving behaviour ($c$), could be expressed as the sum of its direct effect ($c'$) and its indirect effects through each of the $k$ mediator paths [58], which can be expressed as a product of the form $a_i b_i$ (with $i$ denoting the respective mediator path):

$$c = c' + \sum_{i=1}^{k} a_i b_i$$

The direct effect, $c'$, can be interpreted as the effect of risky driving behaviour, controlling for the effects of the assumed mediators, i.e. $b_1$-$b_7$. Significance of indirect effects was assessed using bias-corrected bootstrap confidence intervals (CIs), based on a total of 1000 bootstrap samples. Using the 95% percentile-based level of confidence, indirect effects were judged to be significant when the bias-corrected bootstrap CI for a × b did not contain zero [58]. In order to establish basic relationships between variables, bivariate correlations between the various dimensions of the game experience and risky driving behaviour (WRBTV)were calculated using the Pearson product-moment correlation coefficient.

The value of the mean latency until the button was pressed in the WRBTV was used as the criterion measure (Y), indicative of risky driving behaviour. The scores of the GEQ, collapsed across the three lab sessions within each of the subscales, were used as the mediating variables (M). The experimental condition (*"rule-compliant racing"* games—coded '0'; *"drive'em up"* games—coded '1'.) served as the independent variable (X) influencing the criterion of the regression equation (see also Fig 1).

The analyses were performed using the statistical package IBM SPSS Statistics 23 for Windows with SPSS macro "PROCESS" ([58]; download: http://www.processmacro.org/).

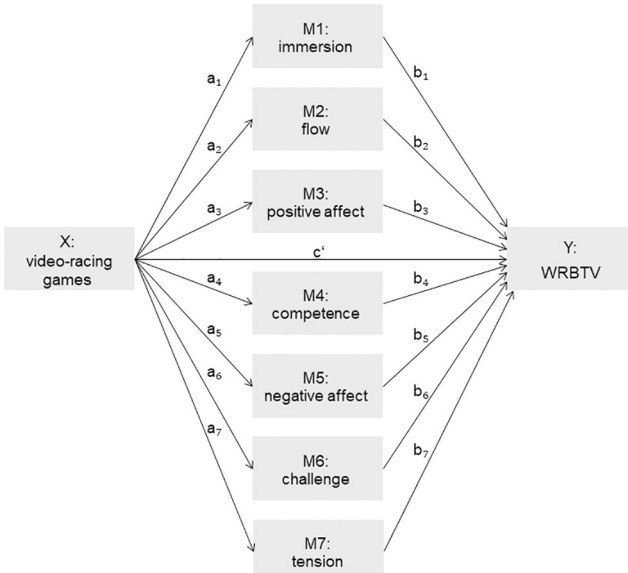

**Fig 1. Conceptual diagram presenting multiple mediator effects of dimensions of the game experience while playing VRGs on risky driving behaviour.**

**Table 1. Descriptive statistics of the study variables.**

|  | N | M | SE | SD | Cronbach's Alpha |
|---|---|---|---|---|---|
| **WRBTV** | 120 | 7.84 | 0.142 | 1.551 | - |
| **GEQ immersion** | 116 | 4.41 | .160 | 1.725 | .835 |
| **GEQ flow** | 113 | 5.99 | .207 | 2.202 | .900 |
| **GEQ positive affect** | 117 | 6.85 | .172 | 1.861 | .864 |
| **GEQ competence** | 119 | 4.94 | .205 | 2.238 | .916 |
| **GEQ negative affect** | 118 | 3.75 | .173 | 1.878 | .847 |
| **GEQ challenge** | 116 | 6.15 | .148 | 1.591 | .694 |
| **GEQ tension** | 115 | 4.50 | .171 | 1.838 | .834 |

N: participant score; M: mean; SE: standard error of the mean; SD: standard deviation.

## Results

### Descriptive statistics

All means, standards errors, standard deviation and Cronbach's alpha, as well as the participants' scores regarding the study variables are shown in Table 1. As can be seen from the table, all of the scales had acceptable-to-excellent internal consistency values well above .700, as assessed by Cronbach's alpha. Therefore, all scales were included in the statistical analyses.

### Correlational findings

Bivariate Spearman's rank correlation analyses examining possible relationships between the various dimensions of the game experience and risky driving were conducted. Analysis results are presented in Table 2. Significant positive correlations were found between *flow* and the other sub-scales, except with *negative affect*, which was significant and negative. The correlations coefficient between *competence* and *immersion*, between *competence* and *flow*, and

**Table 2. Correlation among study variables.**

|  |  | 1. | 2. | 3. | 4. | 5. | 6. | 7. | 8. |
|---|---|---|---|---|---|---|---|---|---|
| **1.** | **WRBTV** | - |  |  |  |  |  |  |  |
| **2.** | **Immersion** | r = .265**<br>n = 116 | - |  |  |  |  |  |  |
| **3.** | **Flow** | r = .078<br>n = 113 | r = .581**<br>n = 110 | - |  |  |  |  |  |
| **4.** | **positive affect** | r = .309**<br>n = 117 | r = .633**<br>n = 115 | r = .423**<br>n = 111 | - |  |  |  |  |
| **5.** | **Competence** | r = .281**<br>n = 119 | r = .415**<br>n = 115 | r = .202*<br>n = 113 | r = .539**<br>n = 116 | - |  |  |  |
| **6.** | **negative affect** | r = -.146<br>n = 118 | r = -.432**<br>n = 114 | r = -.470**<br>n = 111 | r = -.566**<br>n = 115 | r = -.292**<br>n = 117 | - |  |  |
| **7.** | **Challenge** | r = .129<br>n = 116 | r = .562**<br>n = 113 | r = .656**<br>n = 110 | r = .228*<br>n = 114 | r = -.057<br>n = 115 | r = -.269**<br>n = 114 | - |  |
| **8.** | **Tension** | r = .089<br>n = 115 | r = .296**<br>n = 111 | r = .576**<br>n = 108 | r = -.070<br>n = 112 | r = -.195*<br>n = 114 | r = .073<br>n = 113 | r = .636**<br>n = 113 | - |

n-value: the number of persons; r-value: the Spearman's rank correlation coefficients;

*p < .05;

**p < .01.

between *competence* and *positive affect*, were all significant and positive. Correlations between *competence* and *negative affect*, as well as *competence* and *tension* were statistically significant and negative. No significant relationship was found between *competence* and *challenge*. Risky driving behavior (assessed using the WRBTV) was significantly and positively related to *immersion*, *positive affect* and *competence*.

## Mediation analyses

The parallel multiple mediator model provided the opportunity to elucidate the indirect effect of the various dimensions of the game experience, while simultaneously controlling for the other experiential dimensions. Results are shown in Tables 2 and 3.

As can be seen from the left side of Table 2, the experimental manipulation led to significant differences in various dimensions of the game experience. *Drive'em up games* led to significantly increased feelings of *immersion* ($aM_1$), *flow* ($a_2$), *positive affect* ($a_3$) and *competence* ($a_4$), which was associated with a significantly reduced level of *negative affect* ($a_5$). There was no significant effect of game condition for either the *challenge* ($a_6$) or the *tension* ($a_7$) dimension of the GEQ.

In the model of risky driving behaviour (Y), both the paths from *flow* ($b_2$) and *competence* ($b_4$) to risky driving were statistically significant. Direction of effects showed that higher levels of competence feelings led to increased latency values in the WRBTV, indicative of more risk-prone driving behaviour. In contrast, higher levels of flow experience led to reduced latency values in the WRBTV, i.e. less risk-prone driving behaviour. No other dimension of the game experience was significantly associated with the latency values in the WRBTV (see Table 3). Additionally, the direct effect of the experimental condition (X) upon latency values in the WRBTV was insignificant (c' = .242, SE = .358, t = .677, p = .500). This is of importance, since the total effect of the experimental manipulation was significant (c: r = .659, SE = .312, t = 2.112, p = .037). Taken together, these findings imply that the effects of *drive'em up games* on risky driving were fully mediated by at least some of the experiential dimensions assessed by the GEQ. In order to identify the indirect effects of statistical significance, 95% bias-corrected bootstrap confidence intervals for each of the indirect effects $a_i b_i$ served as a statistical test (see Table 4). These analyses revealed that the indirect effects of the experimental condition were significant both for the *flow* ($a_2 b_2$ = -.326, SE = .210, LLCI = -.956, ULCI = -.043) and the *competence* ($a_4 b_4$ = .524, SE = .299, LLCI = .064, ULCI = 1.227) mediator paths. These findings suggest that *drive'em up games* lead to stronger competence feelings, which may, in turn, lead to higher risk driving behaviour. Conversely, those who experienced higher levels of

**Table 3. Direct paths from video-racing games to mediators and direct effect of mediators on risky driving behaviour, examined with the WRBTV.**

| Mediators (M) GEQ | Effects of X upon M (a) | | | | Effects of M upon Y (b) | | | |
|---|---|---|---|---|---|---|---|---|
| | *Coeff* | *SE* | *T* | *p* | *Coeff* | *SE* | *t* | *p* |
| **Immersion** | .710 | .344 | 2.063 | **.042** | -.054 | .149 | -.363 | .717 |
| **Flow** | 1.184 | .425 | 2.787 | **.006** | -.276 | .115 | -2.396 | **.019** |
| **positive affect** | 1.402 | .344 | 4.078 | < **.001** | .080 | .131 | .610 | .543 |
| **Competence** | 2.073 | .388 | 5.338 | < **.001** | .253 | .096 | 2.649 | **.010** |
| **negative affect** | -1.302 | .359 | -3.626 | < **.001** | -.137 | .116 | -1.184 | .239 |
| **Challenge** | .153 | .320 | .477 | .635 | .194 | .159 | 1.223 | .225 |
| **Tension** | -.264 | .369 | -.715 | .476 | .237 | .130 | 1.829 | .071 |

Video-racing games = X; the risk-glorifying racing video games—drive'em up games: experimental group (X = 1); rule-compliant racing video games—racing games: control group (X = 0); risky driving behaviour (WRBTV) = Y; various dimensions of the game experience (GEQ) = M.

**Table 4. Point estimates, standard errors (SE) and 95% confidence intervals for indirect effects of playing video-racing games on risky driving behaviour through various dimensions of the game experience (GEQ).**

| GEQ | Point estimate | Boot SE | Bootstrap 95% CI | |
|---|---|---|---|---|
| | | | *Lower* | *Upper* |
| **Total indirect** | .417 | .252 | -.025 | .955 |
| **Immersion** | -.039 | .122 | -.381 | .154 |
| **Flow** | -.326 | .210 | **-.956***[*] | **-.043**[*] |
| **Positive affect** | .112 | .197 | -.244 | .550 |
| **Competence** | .524 | .299 | **.064**[*] | **1.227**[*] |
| **Negative affect** | .179 | .183 | -.107 | .672 |
| **Challenge** | .030 | .075 | -.077 | .262 |
| **Tension** | -.063 | .106 | -.410 | .074 |

GEQ: various dimensions of the game experience;

[*]CI for a × b did not contain zero.

flow demonstrated less risky behaviour in simulated road traffic situations. None of the other mediator pathways showed significant indirect effects of the racing game condition, with all corresponding CIs containing zero as a value (see Table 4).

## Discussion

The parallel multiple mediator model provided an opportunity to examine the indirect effect of the various dimensions of the game experience, while simultaneously controlling for the other experiential dimensions. The experimental manipulation caused marked differences in dimensions of the game experience. *Drive'em up games* led to significantly increased feelings of *immersion*, *flow*, *positive affect* and *competence*, which was associated with a significantly reduced level of *negative affect*. There was no significant effect of the game condition for either the *challenge* or the *tension* dimension of the GEQ. This suggests that playing risk-promoting video games led to significantly positive cognitive states and emotions. According to Boyle and colleagues [24], the enjoyment experienced during play leads to positive expectations and a positive attitude towards the video game, which, in turn, increases motivation to play.

Assessing the individual dimensions of the video gaming experience examined in their work, Csikszentmihalyi [33] and Sherry [36], for example, postulate that the experience of *flow* creates motivation to play video games, since *flow* is, in itself, rewarding. It can therefore be assumed that individual aspects of the video gaming experience provide a motivation and therefore a reason for the decision to play video games. A high level of motivation to play video games induced by a strong sense of enjoyment derived from the experience could also affect the amount of time spent playing [59]. More frequent gaming for longer periods can influence the individual with what in the GLM are described as personal and situational input variables. Yee [60], for example, shows how the amount of time spent playing can affect human input variables. Vorderer, Klimmt [61] describe further reasons for the consumption of media such as video games in addition to the link to the enjoyment it provides. The authors developed a model in which media-generated enjoyment is the central element [61].

As Vorderer and colleagues [61] surmise, individuals consume certain media for emotion or mood regulation, which is a cause of media-generated enjoyment. In their study, Bowman and Tamborini [62] were able to show that playing a flight simulator game helped regulate the mood of test subjects who had previously reported feeling bored or stressed.

The result related to the gaming experience of challenge is interesting, given the high correlation between *challenge* and *flow*. According to Sherry [36], *flow* arises when there is a balance between *challenge* and the *competence* of the player. Since players reported a significant *flow* experience, the question arises why there was no difference in the *challenge* scale. The significant result for the *flow* scale would then be due to the differing player perceptions of their *competence*. There was a significant difference in perceived *competence*, as shown in this work's *competence* scale. However, our results show a negative correlation between *competence* and *challenge*. It is very likely that the greater the sense of *competence* experienced by the player, the lower the feeling of *challenge*, resulting in a reduction in the intensity of *flow*.

In the model of risky driving behaviour, both the paths from *flow* and *competence* to risky driving were statistically significant. Direction of effects showed that higher levels of competence feelings led to increased latency values in the WRBTV, indicative of more risk-prone driving behaviours. In contrast, higher levels of flow experience led to reduced latency values in the WRBTV, i.e. less risk-prone driving behaviour. No other dimension of the game experience was significantly associated with the latency values in the WRBTV.

Based on the results of the present work, it can be assumed that *flow* and *competence* underlie processes leading to increased risk-taking. The enjoyment derived from playing could influence the personal and situational input variables through motivational processes (e.g. through longer playing sessions). Thus, the game experience could have a long-term impact on video gaming behaviour and its consequences. It can also be assumed that the game experience has a short-term effect on the player's internal state, since the perceived experience of *flow* is a part of those cognitive states, emotions, and excitement that change in the course of playing. It therefore appears appropriate to expand the GLM by adding the important sense of *flow* and *competence* as another possible factor that could influence risk appetite.

Additionally, the direct effect of the experimental condition on latency values in the WRBTV was insignificant. This is of importance, since the total effect of the experimental manipulation was significant. Taken together, these findings imply that the effects on risky driving behavior of *drive'em up games* were fully mediated by at least some of the experiential dimensions assessed by the GEQ. These findings indicate that *drive'em up games* lead to stronger competence feelings, which, in turn, may lead to higher-risk driving behaviour. Conversely, those who experienced relatively higher levels of *flow* demonstrated less risk-taking behaviour in simulated road traffic situation. None of the other mediator pathways showed significant indirect effects of the racing game condition, with all corresponding CIs containing zero as a value.

The correlations between the consumption of risk-glorifying video games and the willingness to take risks in traffic have previously been shown in correlative studies [7, 11, 12, 63, 64]. The drawback of these investigations is that no causal conclusions regarding the effect of video games on risk appetite can be drawn. In addition, there is the question of the direction of effect, since it cannot be clearly demonstrated whether people who are risk-positive in traffic are more likely to have a predilection for daring video games or whether the consumption of video games leads to increased risk-taking on the road. This study used an experimental design in order to allow the drawing of conclusions concerning the consumption of risk-glorifying videogames on actual risk-taking behaviours. Looking at the results of comparable experiments, the significant difference in risk appetite is consistent with that found by Fischer and colleagues [10, 11]. The result of the present work, in contrast to most of the existing research literature, has shown that the experience of playing VRGs lays the foundation for subsequent risk-taking behaviours in traffic. In addition, the effect is not limited to any specific video racing game, since three games were used in the experiment.

It may be surmised, therefore, that the effect of increased risk-taking behaviour, according to the GLM, will reflect the effect of video gaming in the short term. The subjects played video games for a total of one hour over just three days, followed on the third day by a test of their risk appetite in simulated traffic situations using the WRBTV. In accordance with the GLM of Buckley and Anderson [14], each person underwent a triple learning session. The short-term effects measured in this work are, presumably, based on priming processes. By playing risk-promoting video games, scripts and schemata could have been activated via an associative network, which, in turn, may have led to changes in cognition, emotion, and arousal, ultimately affecting risk appetite, as demonstrated in the present work. Playing the games three times could, however, have activated scripts and connections within the network even more strongly and thus made them more accessible. If this were the case, not only short-term effects but also long-term effects according to the GLM would occur. It would therefore be interesting to test whether the changes in risk appetite would appear after only one or two days' playing. If no effect was evident after only one or two days, the changes in behaviour would not be able to be linked to short-term effects, since they would otherwise be detectable immediately after consuming video games. If an effect were to appear only after three days, it would be clear that it had built up over the three days of play, implying that more frequent playing creates learning processes and develops more scripts and schemata that result in a greater propensity for risk taking. It would also be interesting to ascertain whether the effect would also occur after a time delay, e.g. after 24 hours, as considered in the study by Fischer and colleagues [10], where the increased risk appetite, as measured by the WRBTV, was still evident after the delay in testing.

Collectively, our results suggest that the game experience probably directly underlies processes leading to increased risk-taking. Nevertheless, the game experience remains an important factor within the GLM and probably exerts some influences, through motivational processes, on risk-taking behaviour. The results regarding aspects of risk-taking may support findings in previous studies [10], which demonstrated the potential negative effect of *drive'em up games* on driving style and the possible implications of such games for road traffic safety.

## Supporting information

**S1 File. Data set.** SPSS statistics data document.
(SAV)

## Author Contributions

**Conceptualization:** Ewelina Stollberg, Klaus W. Lange.

**Data curation:** Ewelina Stollberg, Klaus W. Lange.

**Formal analysis:** Ewelina Stollberg, Klaus W. Lange.

**Methodology:** Ewelina Stollberg, Klaus W. Lange.

**Supervision:** Klaus W. Lange.

**Visualization:** Ewelina Stollberg.

**Writing – original draft:** Ewelina Stollberg.

**Writing – review & editing:** Klaus W. Lange.

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
