## [Decision Letter · Decision Letter 0]

24 Aug 2020

PONE-D-20-22924

The effects of video racing games on risk-taking in consideration of the game experience

PLOS ONE

Dear Dr. Stollberg,

Thank you for submitting your manuscript to PLOS ONE. After careful consideration, we feel that it has merit but does not fully meet PLOS ONE’s publication criteria as it currently stands. Therefore, we invite you to submit a revised version of the manuscript that addresses the points raised during the review process.

We look forward to receiving your revised manuscript.

Kind regards,

Feng Chen

Academic Editor

PLOS ONE

Journal Requirements:

Reviewers' comments:

Reviewer's Responses to Questions

**Comments to the Author**

1. Is the manuscript technically sound, and do the data support the conclusions?

Reviewer #1: Yes

Reviewer #2: Yes

2. Has the statistical analysis been performed appropriately and rigorously? 

Reviewer #1: Yes

Reviewer #2: Yes

3. Have the authors made all data underlying the findings in their manuscript fully available?

Reviewer #1: Yes

Reviewer #2: Yes

4. Is the manuscript presented in an intelligible fashion and written in standard English?

Reviewer #1: Yes

Reviewer #2: Yes

5. Review Comments to the Author

Reviewer #1: This paper investigates the effects of video racing games on risk-taking in consideration of the game experience. The research topic is interesting and worth of investigation. The experiments are well designed and performed. The authors are suggested to make some minor revisions to further improve the manuscript. The comments are presented as follows:

The Introduction Section is somewhat lengthy. It is suggested to describe the research background more concisely, or to move the part on previous studies into a Literature Review Section.

Some references on correlation test are suggested to add in the text, such as:

A multivariate random parameters Tobit model for analyzing highway crash rate by injury severity. Accident Analysis and Prevention, 2017, 99: 184-191.

Jointly modeling area-level crash rates by severity: A Bayesian multivariate random-parameters spatio-temporal Tobit regression. Transportmetrica A: Transport Science, 2019, 15(2): 1867-1884.

Spatial joint analysis for zonal daytime and nighttime crash frequencies using a Bayesian bivariate conditional autoregressive model. Journal of Transportation Safety and Security, 2020, 12(4): 566-585.

In addition, multicollinearity diagnoses can be also conducted (please refer to the above papers). Significant multicollinearity would have an adverse impact on the estimation results of mediation analysis.

The practical implications of the analysis results should be pointed out.

At end of the text, a Conclusion Section should be added to summarize the findings, to illustrate the limitations, and to provide some directions for future research.

Reviewer #2: The topic of this paper is interesting and important. The methods sound. The results are meaningful and useful. There are several suggestions to improve this paper.

1. There are some typos in this paper. For example, "comprised30 men", "20minutes".

2. The reference style is not so correct. For example, "Fischer, Kubitzki (11)".

3. Is there some information of the driving experience of the participants?

4. For the correlation test, the author could refere to the following paper.

[1] Investigation on the Injury Severity of Drivers in Rear-End Collisions Between Cars Using a Random Parameters Bivariate Ordered Probit Model, International Journal of Environmental Research and Public Health, 2019, 16(14) , 2632.

5. Some references related with driving simulation, maybe are helpful.

[2] Examining the safety of trucks under crosswind at bridge-tunnel section: A driving simulator study, Tunnelling and Underground Space Technology, 2019, 92, 103034. https://doi.org/10.1016/j.tust.2019.103034

[3] Examining the influence of decorated sidewaall in road tunnels using fMRI technology, Tunnelling and Underground Space Technology, Volume 99, 2020, https://doi.org/10.1016/j.tust.2020.103362

6. PLOS authors have the option to publish the peer review history of their article (what does this mean?). If published, this will include your full peer review and any attached files.

Reviewer #1: No

Reviewer #2: No

---

## [Author Response · Author response to Decision Letter 0]

9 Sep 2020

Dear Academic Editor and Reviewers,

Thank you very much for your feedback, comments and suggestions. The responses to your comments are as follows:

Reviewer #1:

Comment 1: The Introduction Section is somewhat lengthy. It is suggested to describe the research background more concisely, or to move the part on previous studies into a Literature Review Section.

# The Introduction has been substantially shortened, and the research background is presented more concisely.

Comment 2: Some references on correlation test are suggested to add in the text, such as: 

A multivariate random parameters Tobit model for analyzing highway crash rate by injury severity. Accident Analysis and Prevention, 2017, 99: 184-191.

Jointly modeling area-level crash rates by severity: A Bayesian multivariate random-parameters spatio-temporal Tobit regression. Transportmetrica A: Transport Science, 2019, 15(2): 1867-1884.

Spatial joint analysis for zonal daytime and nighttime crash frequencies using a Bayesian bivariate conditional autoregressive model. Journal of Transportation Safety and Security, 2020, 12(4): 566-585.

# These references have been added. 

Comment 3: In addition, multicollinearity diagnoses can be also conducted (please refer to the above papers). Significant multicollinearity would have an adverse impact on the estimation results of mediation analysis.

# We attempted to investigate whether it is the game experience that underlies processes leading to increased risk-taking while playing video racing games. A mediation model is one of the best models to identify the mechanism that underlies an observed relationship between an independent and a dependent variable through the inclusion of a third hypothetical variable. This analysis is based on the multiple linear regression model. The multicollinearity analysis, in which one predictor variable in a multiple regression model can be linearly predicted from the others, is in this case not necessary to explain our hypothesis. 

Comment 4: The practical implications of the analysis results should be pointed out. At end of the text, a Conclusion Section should be added to summarize the findings, to illustrate the limitations, and to provide some directions for future research.

# A Conclusion Section has been added. 

Reviewer #2: 

Comment 1: There are some typos in this paper. For example, "comprised30 men", "20minutes".

The typos have been corrected. 

Comment 2: The reference style is not so correct. For example, "Fischer, Kubitzki (11)".

# The reference style has been edited. 

Comment 3: Is there some information of the driving experience of the participants?

# Unfortunately, we have no information on the driving experience of the participants. It is important to note that the participants were randomly divided into two groups. The order of the games was also randomised in both groups. Therefore the statistical analysis has been performed appropriately. 

Comment 4: For the correlation test, the author could refere to the following paper.

[1] Investigation on the Injury Severity of Drivers in Rear-End Collisions Between Cars Using a Random ParametersBivariate Ordered Probit Model, International Journal of Environmental Research and Public Health, 2019, 16(14), 2632.

# This reference has been added. 

Comment 5: Some references related with driving simulation, maybe are helpful.

[2] Examining the safety of trucks under crosswind at bridge-tunnel section: A driving simulator study, Tunnelling and Underground Space Technology, 2019, 92, 103034. https://doi.org/10.1016/j.tust.2019.103034

[3] Examining the influence of decorated sidewaall in road tunnels using fMRI technology, Tunnelling and Underground Space Technology, Volume 99, 2020, https://doi.org/10.1016/j.tust.2020.103362

# These references have been added.

Many thanks again.

Ewelina Stollberg and Klaus W. Lange

---

## [Decision Letter · Decision Letter 1]

25 Sep 2020

The effects of video racing games on risk-taking in consideration of the game experience.

PONE-D-20-22924R1

Dear Dr. Stollberg,

We’re pleased to inform you that your manuscript has been judged scientifically suitable for publication and will be formally accepted for publication once it meets all outstanding technical requirements.

Kind regards,

Feng Chen

Academic Editor

PLOS ONE

Additional Editor Comments (optional):

Reviewers' comments:

Reviewer's Responses to Questions

**Comments to the Author**

1. If the authors have adequately addressed your comments raised in a previous round of review and you feel that this manuscript is now acceptable for publication, you may indicate that here to bypass the “Comments to the Author” section, enter your conflict of interest statement in the “Confidential to Editor” section, and submit your "Accept" recommendation.

Reviewer #1: All comments have been addressed

Reviewer #2: All comments have been addressed

2. Is the manuscript technically sound, and do the data support the conclusions?

Reviewer #1: (No Response)

Reviewer #2: Yes

3. Has the statistical analysis been performed appropriately and rigorously? 

Reviewer #1: (No Response)

Reviewer #2: Yes

4. Have the authors made all data underlying the findings in their manuscript fully available?

Reviewer #1: (No Response)

Reviewer #2: Yes

5. Is the manuscript presented in an intelligible fashion and written in standard English?

Reviewer #1: (No Response)

Reviewer #2: Yes

6. Review Comments to the Author

Reviewer #1: (No Response)

Reviewer #2: (No Response)

7. PLOS authors have the option to publish the peer review history of their article (what does this mean?). If published, this will include your full peer review and any attached files.

Reviewer #1: No

Reviewer #2: No

---

## [Editor Report · Acceptance letter]

30 Sep 2020

PONE-D-20-22924R1 

The effects of video racing games on risk-taking in consideration of the game experience 

Dear Dr. Stollberg:

I'm pleased to inform you that your manuscript has been deemed suitable for publication in PLOS ONE. Congratulations! Your manuscript is now with our production department. 

Kind regards, 

on behalf of

Dr. Feng Chen 

Academic Editor

PLOS ONE